

# Automatic nutrient estimator: distributing nutrient solution in hydroponic plants based on plant growth

Tupili Sangeetha[1] and Ezhumalai Periyathambi[2]

[1] Department of Information Technology, Rajalakshmi Engineering College, Chennai, Tamil Nadu, India
[2] Department of Computer Science and Engineering, R.M.D. Engineering College, Kavaraipettai, Tamil Nadu, India

Corresponding author
Tupili Sangeetha,
tupilisangeetha@gmail.com

## ABSTRACT

**Background:** The primary objective is to address the specific needs of plants at different growth stages by delivering precise nutrient concentrations tailored to their developmental requirements. Challenges such as uneven nutrient distribution, fluctuations in pH and electrical conductivity, and inadequate nutrient delivery pose potential hindrances to achieving optimal plant health and yield in hydroponic systems. By overcoming these challenges, the hydroponic farming community aims to enhance the accuracy of nutrient dosing, streamline automation processes, and minimize resource wastage. Hydroponics, a cultivation technique without soil, facilitates the growth of organic vegetation while concurrently minimizing water use and eliminating the necessity for pesticides. In order to achieve effective cultivation of hydroponic plants, it is essential to maintain a controlled environment that encompasses essential factors such as temperature, carbon dioxide ($CO_2$) levels, oxygen availability, and appropriate lighting conditions. Additionally, it is crucial to ensure the provision of vital nutrients to maximize output and productivity. Due to the demanding nature of a hydroponic farmer's schedule, it is necessary to minimize the amount of time dedicated to nutrient management, as well as pH and EC adjustments.

**Methods:** In order to determine and deliver the proper amount of vital nutrients, such as nitrogen, phosphorus, and potassium, based on the plant growth stage, we presented an automatic hydroponic nutrient estimator in this system. We noticed that the plant's nutrient consumption varies depending on its stage of growth according to plant psychology. Four peristaltic pumps with the necessary sensors are controlled by an Arduino board in the suggested system. Both filling and draining the water are done using each pump. To identify the plant stage, we apply the Plant Growth Stage Identification algorithm to encompass the seedling, vegetative, flowers, and fruit stages.

**Results:** The experimental results reveal that the Growth Stage Identification algorithm obtains 97.5% accuracy for the first 5 weeks with 1,715 ppm of nutrition ingestion, identifying the vegetative state. The flowering stage is identified with 97.5% accuracy in the 6–9th week with 2,380 ppm of nutrition consumption, and the fruiting location is determined with 99.4% accuracy in the last 10–15th week with 2,730 ppm of nutrition consumption.

# INTRODUCTION

The world's population is projected to increase from 6.8 billion today to 9.1 billion by 2050. This growth will necessitate a 25% increase in food production compared to the current levels (*FAO, 2022a*). According to research conducted by the FAO, meeting this demand and simultaneously addressing issues such as hunger, poverty, efficient management of finite natural resources, and adaptation to climate change will pose significant challenges for global agriculture in the upcoming decades. The Food and Agriculture Organization (FAO) (*FAO, 2022b*) projects that in India, more than 189 million people may receive less food in terms of quantity and quality for healthy growth by 2020. In India, the population is increasing rapidly, but at the same time, the percentage of agricultural land is decreasing. At the same time, inadequate water resources, fewer rainfalls, increased expenses, and sudden disasters decreased farmers' interest in farming. As a result, many of them have transitioned into the real estate sector to seek higher profits within a shorter timeframe (*Shirsekar et al., 2021*).

Hydroponics is a branch of hydroculture that is the most suitable explanation for these difficulties. Hydroponics farming (*Tagle et al., 2018*) is an intelligent way of growing crops that yields high-quality organic plants in a limited space without using soil and does not expose them to external disasters, and not causing any environmental pollution. The hydroponic food production method is becoming more popular for cultivating crops, especially for urban people who can grow plants on balconies or rooftops (*Kaewwiset & Yooyativong, 2017*).

The hydroponic food production method of cultivation, referred to as "soilless culture" (*Resh, 2022*), uses water or nutrient solutions as the growing standard.

Possibility of using land that is unsuitable for conventional agriculture, such as dry and degraded terrain. The ability of plants to adapt to shifting weather patterns. Cultivation is possible all year. Reduced need for labor-intensive operations such as weeding and soil preparation. These are the method's primary advantages (*Gumisiriza et al., 2022*).

There is debate about whether hydroponic farming is more active than conventional farming (*Atikah & Widyawati, 2021*; *Lee & Lee, 2015*). Hydroponics agriculture has become a revolution in many countries and is slowly spreading in India. It has proved that it can ultimately grow pesticide-free crops with 95% less water and consume reduced nutrients compared to conventional agriculture.

Six different methods of growing hydroponics include deep water culture, wick system, nutrient film technology, ebb and flow systems, drip systems, and aeroponics. It can choose any growing medium except soil like coconut pellets, perlite gravel, rockwool, or sand, or it can be any nutrient explanation. Plants are grown either horizontally or vertically based on space availability (*Nursyahid et al., 2021*). In our system, we used deep water culture to grow plants, as shown in Fig. 1.

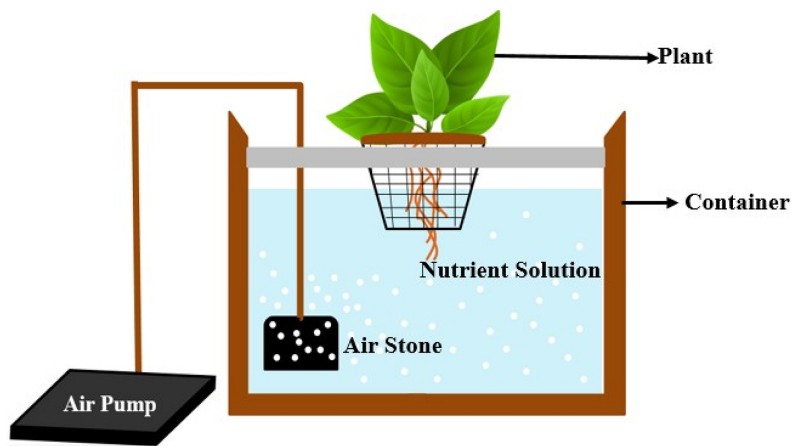

**Figure 1  Deep water culture for hydroponics.**

The entire method for a vertical farming controlled environment comprises three significant steps. The first step is about intelligent sensors for monitoring several parameters, including air density, humidity, light intensity, temperature, $CO_2$ levels, oxygen levels, water inflows, and recording pH and EC values. The second step covers caring for the plants under a controlled environment and regulating all the above parameters in specific ranges or threshold values. The last step is crucial for monitoring plant disease detection, weeds detection, pollination, prediction of nutrients mixture along with plant growth maturity stage identification, and messaging complete crop information through smart phone to inform the farmer about changes and for good yielding (*Saputra, Irawan & Nugraha, 2017*).

The fundamental factors which affect a plant's growth include pH, temperature, $CO_2$, and EC of the water in the system. Four primary sensors control the environment in our hydroponics system. They are DHT11 (temperature and humidity), ultrasonic sensors (water flows), EC sensors (nutrient dissolvability), and pH sensors (acidity of nutrient solution). For these factors, we use an Arduino UNO to communicate and monitor whether or not the readings are in the given threshold range.

Instead of soil, hydroponics is fed with nutrient solution added to the normal RO water. Hydroponics needs a varying number of macronutrients and micronutrients throughout its life cycle. Macronutrients are called primary nutrients and are crucial for the plant's health; these include nitrogen (N), phosphorus (P), and potassium (K), which are available from Ca $(NO_3)_2$ (calcium nitrate), $KNO_3$ (potassium nitrate), and $KH_2PO_4$ (mono-potassium phosphate). In addition to the NPK mixture, micronutrients are also needed for plants that, include calcium (Ca), manganese (Mn), zinc (Zn), boron (B), iron (Fe), molybdate (Mo), chlorine (Cl), magnesium (Mg), copper (Cu) and sulfur (S). Proper nutrients, like nitrogen, phosphorous, potassium, *etc.*, are essential for plant development. Estimating ratios of NPK solution is a significant step that helps in high productivity (*Shirsekar et al., 2021*).

In hydroponic systems, plant growth and efficiency mainly depend on two main features, EC and pH values, that regulate plant nutrient uptake. We use a pH meter to measure the pH of the nutrient solution. If the pH value measured below seven, that shows acidic, whereas above seven shows basis or alkaline of explanation. The threshold range of pH values of hydroponics plants is mostly from 5.5 to 6.4. The pH value above or below the predetermined threshold can cause nutrient deficiencies that reduce productivity (*Kaewwiset & Yooyativong, 2017*). The electrical conductivity (EC) is proportional to the number of ions dissolved in the explanation. It indicates to the farmer the number of nutrients consumed and still available in the water tank (*Eridani, Wardhani & Widianto, 2017*). The appropriate threshold range is from 2 to 5. The measured EC value determines adding the additional nutrients or reduce the solution with water. If the plants experience excess nutrients, then it becomes poisonous. Even though each nutrient helps plant productivity in all stages, its extra or absence can cause deficiency or toxicity symptoms (*Domingues et al., 2012*).

However, nutrient management efficiency depends on the design and distribution of nutrients considering the plant's maturity and demands. Currently, some developed auto nutrient dosing machines are in the market for feeding hydroponics. In our article, "Related Survey" discussed hydroponic culture and how nutrition is supplied without human intervention. "Proposed System and System Design" visualized the system functionality with a neat block diagram. The procedure for nutrition estimation according to plant stage and demands is described in "Methodology". Experimental results and app details are shown in "Experiment Results".

## RELATED SURVEY

Finding crop growth rate: According to authors (*Verma & Gawade, 2021*), the tomato plant was monitored for 167 days and proposed a system to find the total crop growth rate (CGR). They used a formula to calculate absolute CGR,

*Absolute CGR = w2 − w1/t2 − t1*

W1 and W2 are the dry weights of tomato plants in grams to the time in days t1 and t2. Here plants' needs were fulfilled by distributing nutrient solutions at regular intervals, ensuring the correct amount of chemical composition, and the total 167 days of tomato plants were divided into three stages. At stage I, after 56 days, the absolute CGR of the tomato plant was 2.7 grs/day. At stage II, after 77 days, a nutrient solution is multiplied, and similarly, at 93 days, the absolute CGR is increased to 4.27 grams/day. At stage III, after 116 days, the tomato fruits were matured and harvested to reduce the distribution of nutrient solution with a decrease in CGR values monitored. In this article, nutrient solution ratios are determined and supplied manually.

On or Off controller to control nutrient tank with mix A and B:

*Fadillah, Faroqi & Kamelia (2021)* about AB mix, which is a mixture of Fertilizer A (nitrogen, potassium) and Fertilizer B contains phosphate. They use the On-Off control method. This system receives input from the user to switch on and off the device to supply nutrients from tank A and tank B. This article uses the ESP32 microcontroller to turn on or

off the tank with nutrient mixtures A and B based on the plant height they measured manually.

Formula based Automatic distribution of NPK solution:

*Shirsekar et al. (2021)* discussed controlling and adjusting the distribution of NPK solution and oxygen concentration for hydroponics. They have proposed and used two necessary sensors: NPK sensor and oxygen sensor. They used separate containers to fill the NPK solution, and for oxygen, they were immersed with four pipes to dissipate the solution to a hydroponic tank with fresh RO water. The NPK sensors and oxygen sensor helps in taking live readings of NPK solution and oxygen concentration. These readings were sent to the system, which is with Arduino mega microcontroller, to control the functionality of the pump and thus to adjust the NPK solution in the main hydroponic water tank. Initially, the user set the threshold values of NPK and oxygen. The procedure used in this system mainly calculates alteration among threshold values and actual values to control NPK pump turn ON or OFF. Similarly, if the oxygen concentration required is a minimum of 5 mg/lit and is within the threshold range, then the oxygen pump is OFF, or else required to turn ON the air pump. Using the formula, they calculated the time needed for the NPK pump to dissipate the nutrient solution.

$$Time(sec) = \frac{(n \times T - A) \times V}{F \times C}$$

where

  T = Threshold value of N or P or K solution
  N = Number of plants
  A = Actual values of N or P or K solution
  V = Amount of water tank in liters
  F = Pump flow rate
  C = Nutrient concentration

But here, they did not discuss plant age, the stage at which it has grown, or the amount it needs to be based on plant maturity.

In *Saputra, Irawan & Nugraha (2017)*, the authors proposed a forward chaining technique to define the condition of the nutrient solution for distribution to plants. They used four sensors for measuring temperature, pH, EC, and water levels and seven pumps for solution intake and draining. Under the knowledge base design mentioned in Tables 1 and 2, they said that the system could be in any of the 18 conditions K001 to K018, based on pH and EC measurement conditions from G001 to G008. Table 3 shows a mapping between measurement conditions and the nutrient condition for deciding to switch on or off the pumps. A decision tree was constructed based on the knowledge base collected from the condition mappings. This tree allows the implementation of any 18 nutrient circumstances based on the condition output. The amount in the decision tree represents the nutrient condition. If the condition is true, the search continues from one stage to another along with the branch of the tree and finally gives the right decision to take to supply the right amount of nutrients. For example, if sensor measurements are pH = 6.1,

**Table 1 Survey for automatic hydroponic nutrient estimator.**

| Reference | Objective | Pros | Cons |
|---|---|---|---|
| *Khaoula et al. (2021)* | ML-based nutrient supply optimization | Vernier sensors are connected to a Raspberry Pi to measure nutrients. | There needs to be more information available regarding data storage or transmission. |
| *Gao, He & Yue (2018)* | To regulate and keep track of environmental and water quality factors | A solar-based power source powers node. | The sensor node's Wi-Fi implementation increases power consumption and necessitates more Internet infrastructure. |
| *Arvind et al. (2020)* | Hydroponics system automation using hydroponics and machine learning (ML) edge computing technique | Implemented cloud server to display sensor data | It is only used on a small scale. |
| *Valiente et al. (2018)* | Systems for remote monitoring that use TDS sensors | TDS sensors are subjected to calibration. | The IoT-based system is missing a cloud server. |

**Table 2 Nutrient consumption measured.**

| Week number | pH measured | Average ppm measured/Week (EC X 700) | Stage identified |
|---|---|---|---|
| Week 1 | 5.6 | 1,120 | Vegetative |
| Week 2 | 5.8 | 1,330 | Vegetative |
| Week 3 | 5.5 | 1,470 | Vegetative |
| Week 4 | 5.8 | 1,575 | Vegetative |
| Week 5 | 5.6 | 1,715 | Vegetative |
| Week 6 | 5.4 | 2,100 | Flower |
| Week 7 | 5.6 | 2,240 | Flower |
| Week 8 | 5.5 | 2,250 | Flower |
| Week 9 | 5.7 | 2,380 | Flower |
| Week 10 | 5.5 | 2,390 | Fruit |
| Week 11 | 5.6 | 2,520 | Fruit |
| Week 12 | 5.3 | 2,555 | Fruit |
| Week 13 | 5.2 | 2,450 | Fruit |
| Week 14 | 5.5 | 2,660 | Fruit |
| Week 15 | 5.4 | 2,730 | Fruit |

**Table 3 Accuracy analysis.**

| Week number | Stage identified | CNN | DNN | Proposed |
|---|---|---|---|---|
| Week 1–5 | Vegetative | 95.5 | 94.3 | 97.5 |
| Week 6–9 | Flower | 95.7 | 96.7 | 97.5 |
| Week 10–15 | Fruit | 96.8 | 96.6 | 99.4 |

ppm = 629 and temp = 25 then the right action list includes {pump1 = off, pump2 = off, pump3 = off, pump4 = off, pump5 = off and Fan = off}.

*Tagle et al. (2018)* created a hydroponic tower for urban farming. It was outfitted with various sensor modules to track growth parameters like water temperature and level, total

dissolved solids, relative humidity, ambient temperature, and light intensity. A subsequent investigation has included an automated data acquisition system (DAQ) (*Shetty, Pai & Mallya, 2021*). The DAQ program, written in Arduino software, was intended to collect and store data from six hydroponic parameters.

*Tembe, Khan & Acharekar (2018)* suggested a hydroponics technique in 2018 to meet humans' food requirements. They put the pH probe module and the light spectrum for hydroponics in place. The hydroponics idea can be applied in areas that are experiencing drought. Hydroponic plants can produce a higher yield and quality with less water consumption. It is not affected by the environment in the hydroponics system, and crops can be grown all year. The foremost disadvantage of a hydroponic system is the higher initial setup cost and the requirement to implement automatic monitoring equipment. Likewise, it requires constant supervision from the farmer. Power outages must be resolved manually in *Sihombing et al. (2018)* recently established hydroponic organization that only procedures the temperature of the nutrient clarification and the water level. A smartphone can be used to monitor the system.

The method developed by *Belhekar et al. (2018)* and *Jaimes et al. (2019)* measures the system's relative humidity, temperature, electrical conductivity, pH, and water level. Farmers are alerted to situations like systemic nutrient imbalances by the data submitted to a database. Monitoring the hydroponic atmosphere is the only function of the system. Therefore, the system is without actuators and depends on the farmer to control occurrences (*Belhekar et al., 2018*). While *Jaimes et al. (2019)* can add rainwater to the design and regulate pH levels.

The volume and velocity problem of data generated in a hydroponic system is addressed by *Chaiwongsai (2019)* using data fusion. Data fusion is a technique that merges (fuses) information from many sensors to provide a more accurate result. Moreover, data fusion lessens network transfers, which lowers traffic and energy use. The authors suggested a control and management system for tropical hydroponic horticulture. The system keeps track of data on the state of the power, temperature, humidity, water level, pH, and EC, which are then sent over the internet to a remote database. Users can access a mobile application to verify the hydroponic sensor's status. A two-tier network of sensors (including data fusion and SI sensors) makes up the system (DFi). The data fusion sensors receive the sensed data from the sensors and combine it to create an ideal outcome before saving it in the remote database.

An actuator system was developed employing Vernier sensors to deliver the nutrition into the environment in a closed loop after the nutrient values had been established. To rank the features, this feature selection approach used a tree classifier, the XG Boost classifier, and recursive feature deletion with extra (*Muneeb, Ko & Park, 2021*). To automatically assess the input and produce the appropriate outputs, fuzzy logic is used to construct an aquaponics monitoring and control system. The PDF and FPDF controllers optimize parameters using a genetic method. Compared to the conventional PDF controller, better consequences were obtained in the greenhouse's humidity and temperature management (*Mpofu et al., 2020*). The optimum control of greenhouse temperature is obtained by utilizing a Q-learning strategy to manage greenhouse elements

that are dependent on the environment (*Wang & Lee, 2020*). The survey for the Automatic Hydroponic Nutrient Estimator is shown in Table 1.

*Nemali (2022)*: The oldest known instances of greenhouse use date to the Roman era around 14 CE, but commercial use did not start in the Netherlands until the 20th century. The Netherlands' greenhouses suffered significant damage during World War II, and during the post-war reconstruction efforts, engineers in the Venlo region created the tall, glass, multi-span greenhouse that is commonly seen today. The Netherlands made major post-war efforts to guarantee food security, which included building several greenhouses in the Venlo style for the cultivation of vegetables. Global greenhouse consumption has increased since commercial CEA production was first introduced in the Netherlands. Comparing greenhouse farming to field agriculture, the yield potential has improved significantly due to significant advancements in glass materials, lighting sources, and growth systems. The development of hydroponic agricultural cultivation was one important breakthrough. *Haider & Rai (2021)* The major issue is that our current crop productivity is declining as a result of climate change. It is predicted that major reductions in staple crop production, such as maize (20–45%), wheat (5–50%), and rice (20–30%), will result from rising global temperatures and erratic weather patterns. Moreover, as temperatures rise, up to 30% of annual agricultural production may be lost as a result of increasing phytopathogen and pest pressure.

According to *Bafdal, Dwiratna & Kendarto (2017)*, this technology is an advancement over earlier ones. Smart watering is the system's registered trademark. The valve for the proposed self-aware fertigation is built to adhere to gravity and pressure laws. It implies that the valve will automatically open and shut depending on the predetermined height, enabling adequate nutrient circulation. This kind of technology is excellent for use in places without electricity.

A hydroponic intelligent plant care system that uses IoT technology to control the environment was created by *Paulchamy et al. (2018)*. IoT allows users to make changes by quickly adding or removing sensors and actuators. By managing the water sprinkling and pump flow, the IPCH box effectively reduces the $CO_2$ concentration. According to the level, the flow of nutrients to hydroponic plants is automatically increased. The challenging step is maintaining a steady pH level in the water solvent. The crop can be grown anywhere in hydroponics, where the growth rate is 90% more than in regular farming.

The novelty of the proposed method for distributing nutrient solution in hydroponic plants lies in its groundbreaking approach to optimizing plant growth through advanced modeling techniques. Unlike existing convolutional neural network (CNN) and deep neural network (DNN) models, this innovative system leverages a unique combination of data-driven insights and real-time feedback mechanisms to achieve unparalleled accuracy. The model's claim to surpass existing CNN and DNN models is rooted in its ability to adapt and learn dynamically from the intricate interplay of variables affecting plant nutrition. By integrating cutting-edge sensor technologies and machine learning algorithms, the system not only precisely distributes nutrient solutions tailored to the specific needs of each plant but also continuously refines its predictions based on live environmental data. This adaptive and responsive nature sets it apart, positioning it as a

pioneering solution in the realm of hydroponic agriculture, promising enhanced crop yields and resource efficiency.

# PROPOSED SYSTEM AND SYSTEM DESIGN

The ratios of nutrients will keep changing due to plant type, plant stage, plant organ focus, air temperature, and humidity. Automatic hydroponic nutrient management must focus on-

- Taking complete control of hydroponics that monitor and regulate everything needed for plants, including nutrients, EC, pH, $CO_2$, temperature, and humidity
- Estimating nutrient ratios for feeding plants.
- Automating nutrient supply to dispatch nutrients in the right amounts at the right time.
- Maintain an app that communicates with the farmer and allows remote access to the growing environment.
- Immediate alerts to the farmer for uncommon incidences in the growing environment.
- Use a cloud server to maintain day-to-day plant growth status.

## Block diagram

Hydroponics is soilless cultivation that can be grown in any small indoor area. Each requirement must be provided artificially in a controlled environment, whereas everything is naturally available for soil agriculture. The above Fig. 2 depicts the whole idea of our proposed system.

Our proposed system aims at developing an auto-dosing system for nutrient management to distribute essential nutrients for plants with less human intervention. The system mainly automates the nutrient supply to prevent under-dosing or overdose. This system calculates and estimates the number of nutrients based on the ratios given at every plant stage or age. The block diagram of the system mainly comprises a growing chamber, water reservoir, NPK mixture pumps, and Arduino microcontroller that connects various sensors to sense different parameters like pH, EC, temperature, and water levels and uses LCD $16 \times 2$ monitor to display all these parameters.

The pH and EC sensors in the system continuously monitor the hydroponic solution. If the recorded pH falls beyond the required range, the system triggers pH adjustment processes, such as introducing pH-up or pH-down solutions, to return the pH to the ideal range. If the EC values exceed the predefined thresholds, the system adjusts the nutrient content by diluting the solution with water or supplementing it with more nutrients. These corrective steps are frequently automated and initiated in real-time to handle any variations as quickly as possible and to guarantee that the nutrient solution remains within the precise parameters conducive to plant health. The system provides a stable and controlled environment by proactively maintaining pH and EC levels, protecting against harmful impacts on nutrient uptake and overall plant development in hydroponic farming. It feels pH and EC in regular intervals and compares recorded values with the minimum threshold already initialized in the system. EC and pH ranges vary from plant to plant. The

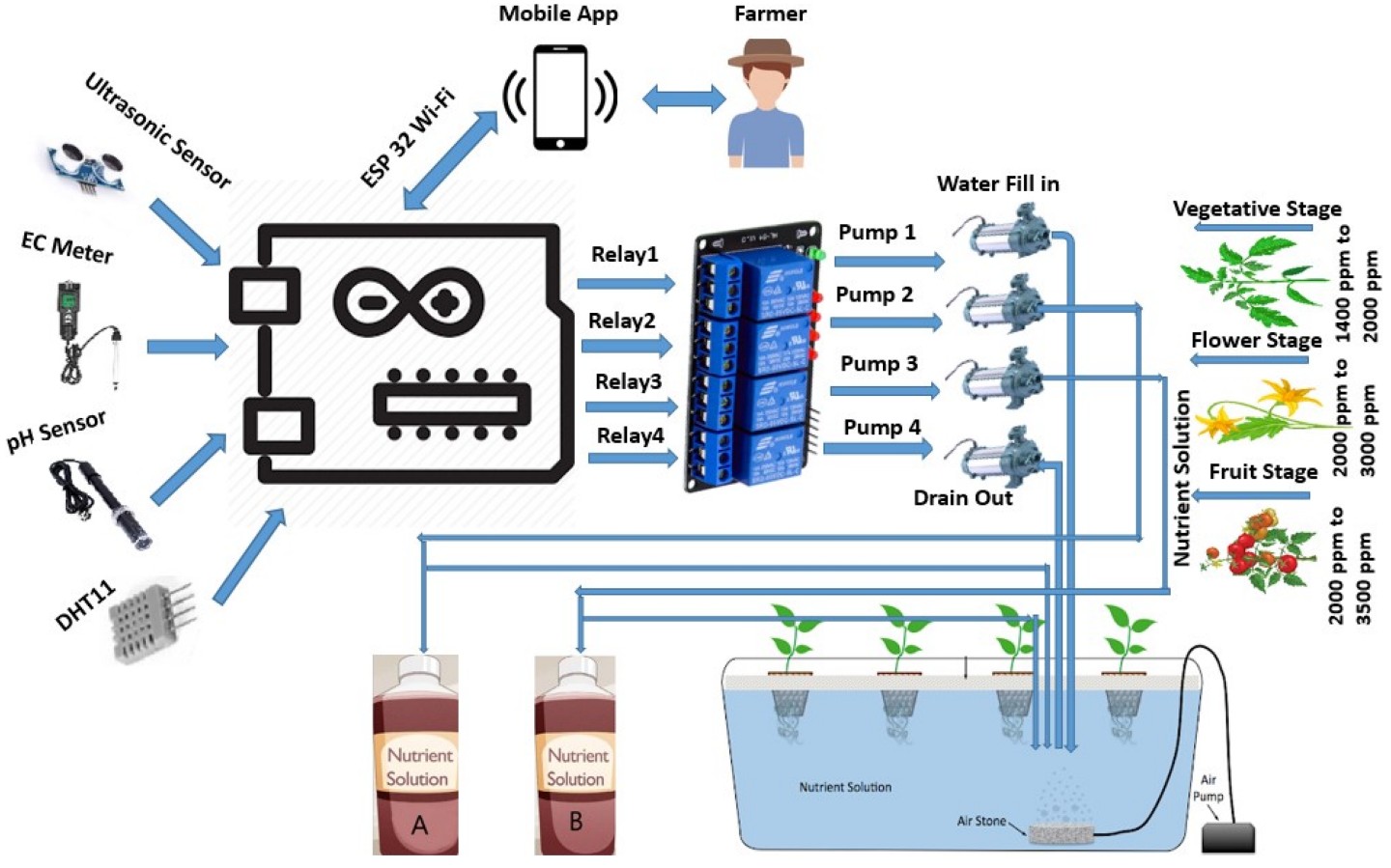

**Figure 2** Block diagram for proposed automatic nutrient provider for hydroponics farming.

nutrient solution requirement of the plant is directly dependent on the growth stage of the plant, which causes EC ranges to go for different growth stages. If the observed pH value is outside the predefined range, remedial measures are being taken to restore a workable pH range. As well as it is checking whether the recorded EC value is higher, lower, or feasible. If recorded EC values are higher, switch on the water pump to reduce the nutrient concentration. On the other hand, if ppm goes down, change on NPK mixture pumps to increase the nutrients in the reservoir.

The Growth Stage Identification algorithm is a sophisticated mechanism that determines the developmental phases of plants in a hydroponic environment, ensuring exact nutrient administration adapted to each growth stage. This program uses multiple data sources to make informed conclusions about the plant's age. Environmental sensors, such as those that measure temperature, humidity, and light intensity, provide real-time information on growing conditions. Furthermore, nutrient sensors monitor the composition of the hydroponic solution, providing information about the plant's nutritional requirements. This abundance of data is processed by the algorithm, which then applies programmed growth patterns appropriate to the farmed plant species. These

patterns, which range from germination to vegetative growth, blooming, and fruiting, serve as a guide for determining the present stage of growth. Machine learning techniques are used in some implementations to improve the algorithm's capabilities by learning from prior data and adjusting to changing environmental conditions. The decision-making mechanism of the algorithm causes the dosing system to distribute the correct amount and composition of nutrients based on the recognized growth stage. In hydroponic farming practices, this dynamic and adaptable technique guarantees that plants receive personalized nourishment throughout their lifecycle, optimizing their health and productivity. While supplying nutrients, the system executes the Growth Stage Identification algorithm to identify the age of the plant and provide the correct quantity of nutrients accordingly. NPK pumps continue flowing nutrient solution for the period calculated in Eqs. (1), (2), or (3) according to the plant stage.

The View at Site app connects the autonomous dosing system to the farmer's mobile smartphone, allowing rapid access to important information. The IoTCloudData cloud server collects and processes data from the hydroponic setup as a centralized hub. The farmer may remotely monitor critical variables such as nutrient levels, pH, and environmental conditions *via* the app. The system sends data on these parameters to the cloud server in real time, and the View at Site app retrieves it and displays it on the farmer's mobile device in an easily understandable way. Furthermore, the app allows the farmer to set and alter dosing parameters, receive notifications when conditions deviate from ideal, and even remotely manage the dosing system. This use of IoT technology not only streamlines the monitoring process, but also provides farmers with the freedom and convenience to operate their hydroponic systems efficiently and proactively, hence improving the precision and productivity of their farming practices. The hydroponic system ensures a dynamic understanding of the plant's immediate surroundings by tracking these parameters in real-time. This information is then processed by algorithms or control systems that use established growth patterns for the individual plant type. The system can properly estimate the current growth stage of the plants by aligning the observed parameters with these growth patterns. This data directs the automated dosing system to distribute the precise amount and composition of nutrients required at each developmental stage, encouraging strong plant health and, ultimately, maximizing potential production in hydroponic farming.

### Circuit diagram
The components of the circuitry proposed system have shown in File S1, including

- Step down transformer for power supply
- Arduino UNO
- Relay board with submersible pumps for pumping water and nutrients
- pH sensor
- EC sensor
- DHT11 sensor for measuring temperature
- Ultrasonic sensor

- 20 × 4 LCD
- MCP3008 10-bit ADC

i) **DHT11 sensor:** DHT11 low-cost digital sensor used in growing hydroponics to measure air temperature and humidity. It usually operates at 3.5 to 5.5 V and can measure temperatures up to 500 C.

ii) **Arduino UNO:** A microcontroller used to connect input-output devices, sensors, and other required electrical devices. It supports the IDE environment to a program to the system with any high-level languages or packages like Python. The power supply with 5 Volts is used to supply power to the board and all the sensors.

iii) **Relays pumping water and nutrients:** A relay is an electrical device that behaves like a switch for pumping water and nutrients into our system. It allows you to switch on or off pumps in regular intervals effortlessly. Totally four relays used in our approach to control water, nutrients, and pH solutions. Each relay has three terminals at the low current side, namely VCC, GND, and IN pins, and three at the high current side NO, C, and NC. The IN pin, connected to Pin 7 in the Arduino board, causes the relay to switch the pump ON or OFF.

iv) **pH meter:** pH is the potential of hydrogen that tells how well plants absorb nutrients. The pH meter calibrates the acidity or alkalinity of nutrient explanation in the water tank. In this system, we have grown tomato plants for the experimental results, and tomato plants can grow at a 5.5 to 6.5 feasible pH range. At all times, pH should be checked after getting EC in the predetermined range. The pH is too low or too high can stop nutrient uptaking and causes some deficiencies in plants. We should use pH up or down solutions to bring into the correct pH ranges.

v) **EC meter:** EC meter and pH meter supports feeding hydro plants nutrients at consistent levels. Electrical conductivity (EC) can be measured in millisiemens (mS) or parts per million (ppm). For tomato plants, optimal EC ranges can be from 2 to 5 mS or 1,400 to 3,500 ppm levels based on the type of crops and growth stage of the tomato plants. If the EC value decreases, plants consume soundly and switch ON NPK pumps to raise nutrient solution. If the solution's EC goes up, it means more nutrients in the resolution, and switching to the ON water pump dilutes the solution.

vi) **Ultrasonic sensor:** An ultrasonic sensor (HC SR04) is used to regulate the water level in the water tank. It is used to degree distance from one object to another object. It controls the water pump to turn on or off, thus avoiding water wastage. If the water equal recorded is less than the inception, it prompts the water pump to turn ON; otherwise, it will turn OFF immediately.

vii) **20 × 4 LCD:** We used 20 columns × 4 rows of character LCD monitor designed for Arduino microcontroller with 16 pins to display all the input from sensors.

viii) **ATMEGA328 ADC:** This device used for analog to digital conversion and *vice versa*.

# METHODOLOGY

We are using deep water culture for growing plants. Submersible pumps are used to mix the nutrient solution into the water reservoir. Hydroponic plants are soilless and consume macro and micro nutrients for their growth. It is essential to remember that the number of hydroponic nutrients and ratios that plants often require increases as plants get grown up. Then they slowly decrease as the plant enters the reproduction or harvesting stage. An auto-dosing system, designed to deliver nutrients mainly to concentrate on the plat stage, should fix specific nutrient ratios accordingly. Let us see how tomato plants consume nutrients at different stages of their growth.

## Tomato plant at germination stage

At the time of the seedling or germination stage, the plant supplied nutrients naturally available in the RO water. The seedling can start its growth with lightening, $CO_2$, oxygen, and minimal nutrients.

## Tomato plant at vegetative stage

N-P-K ratios represent the percentage of each nutrient in the solution. For example, the N-P-K ratio 7-9-5 is the best ratio at the vegetative stage that contains 7% of nitrogen, 9% of phosphorous, and 5% of potassium. The remaining solution is filled with varying amounts of micronutrients.

During the vegetative stage, plants consume higher amounts of nitrogen in the fertilizer for a rapid growth rate. Nutrient consumption with more nitrogen products makes plants healthier, with leaves seen dark shined greenish. At this stage, plants take around 1,400 to 2,000 ppm of nutrients required for growth. In our system, the minimum threshold of 1,400 ppm and a maximum of 2,000 ppm are taken for nutrient estimation.

## Tomato plant at reproductive stage (flower/fruit stage)

Plants' reproduction stage starts with flowers that carry out reproductive functions to produce fruits. During this stage, plants consume higher amounts of phosphorous (P) for flower/fruit production and reduce taking nitrogen as it is already collected in vegetative stage at higher levels. Phosphorous is also needed for healthy root structure development. In this stage, potassium is a crucial nutrient for reproduction. In this stage, a 5-15-14 NPK ratio may be a wise choice for plant growth. Plants require around 2,000 to 4,500 ppm of nutrients for optimum growth throughout the blooming or fruiting periods. In our system, the minimum threshold of 2,000 ppm and a maximum threshold of 2,500 ppm has taken. Similarly, for the fruiting stage, the minimum threshold of 2,200 ppm and ultimate point of 3,500 ppm has been accepted for nutrient estimation. Achieving 97.5% accuracy in recognizing the vegetative stage, 97.5% in the blooming stage, and 99.4% in the fruiting stage in hydroponic plants using nutrient solutions has significant implications for precision agriculture. This level of precision enables focused and efficient fertilizer delivery throughout the plant's growth cycle, optimizing resource efficiency and minimizing waste. With the potential to improve crop health and productivity, such accuracy contributes to the automation of hydroponic systems, decreasing labor requirements and enabling for

data-driven decision-making. Furthermore, the environmental benefits, such as reduced nutrient runoff and a smaller ecological imprint, correspond with sustainable farming practices. This high level of precision not only ensures crop consistency, but also presents hydroponic farming as a competitive and environmentally aware strategy in modern agriculture.

## Equations for nutrients estimation based on the stage

Estimating and calculating the ratios of nutrients to add to nutrient solutions is a primary issue in the hydroponic controlled environment. In India, the hydroponics culture is beginning. Most urban people are aware of growing hydroponics. In this urban farming, a farmer is tightly engaged with his day schedules. They cannot be time-intensive in dosing plants and making pH or EC adjustments. And if he is new to farming, he may accidentally shock his plants by over- or under-dosing.

Overdosing on nutrients can result in a nutritional burn, the creation of salt content, and even plant death. Underdosing of nutrients sometimes results in plant decreases. The use of automated dosing systems is the solution to this problem. This system is used to avoid human errors and simplify nutrient management with less human intervention (dosing, checking levels, measuring pH and EC).

Normally EC is measured in millisiemens. We can convert this EC to ppm using 500 NaCl or 700 KCl scales.

$$1 \, MS = 500 \, ppm = EC \, 500 \quad (or)$$
$$1 \, MS = 700 \, ppm = EC \times 700$$

For the tomato plant, the optimal EC range is from 2 to 5 mS, *i.e.*, 1,400 to 3,500 ppm, according to the KCl scale. We already discussed that plant absorbing nutrients involves several factors such as on plant stage, type of plant, and plant part growing. Accordingly, nutrient consumption at the vegetative stage is different from the flower stage, and similarly, nutrient uptaking at the flower stage is different from the fruit stage. At the germination stage, the plant uses a tiny percentage of nutrients as it can grow with nutrients available in fresh RO water. When it has grown in 30–40 days, leaves increase with height, around 20 to 30 cm. This stage is called the early vegetative stage. The plant expenses around 1,400 to 2,000 ppm of mixed macro and micronutrients for its happy growth. In this entire stage, the plant consumes more nitrogen with around a 7-9-5 NPK ratio. That means the tomato plant takes 7% of nitrogen, 9% of phosphorous, and 5% of potassium out of the total nutrients provided. Our system continuously measures pH and EC values in a growing environment. If EC measured is less than the minimum threshold at the vegetative stage, then the system immediately switches on NPK pumps for the time Tveg micro seconds.

$$Tveg = \frac{C \, X \, (MAXVEGppm - SURppm)}{FR} \tag{1}$$

Here, C is the capacity of the nutrient reservoir. MAXVEGppm is the maximum limit of nutrients needed for tomato plants at the vegetative stage, SURppm is surplus ppm of nutrients available in the nutrient reservoir. FR is flowing pump rate. Next to the vegetative stage, plants begin reproduction and flowering in 40–65 days, called the flower stage. In this stage, the plant expenses around 2,000 to 4,500 ppm of mixed macro and micronutrients for healthy growth. In this stage, the plant consumes less nitrogen but more phosphorous and potassium with around a 5-15-14 NPK ratio. That means the tomato plant takes 5% of nitrogen, 15% of phosphorous, and 14% of potassium out of the total nutrients provided. When the system finds an EC range below the minimum threshold, it is necessary to switch on NPK pumps for Tflow ms. The same calculation is done at the fruiting stage with a point change.

$$Tflow = \frac{C \; X \; (MAXFLOWppm \; - \; SURppm)}{FR} \qquad (2)$$

$$Tfru = \frac{C \; X \; (MAXFRUppm \; - \; SURppm)}{FR} \qquad (3)$$

Here, C is capacity of nutrient reservoir.
MAXFLOWppm is the maximum limit of nutrients needed for tomato plants at the flowering stage,
MAXFRUppm is the maximum limit of nutrients needed for tomato plants at the fruiting stage,
SURppm is surplus ppm of nutrients available in the nutrient reservoir.
FR is flowing pump rate.
The tomato plants start to fruit. It is a very crucial stage, and constant attention is needed. In this Fruit stage, plants' nutrient consumption equals the flowering stage. When EC recordings are not optimal, then it is required to switch on NPK pumps for the time Tfru.

## Automatic nutrient dosing system

Anything that causes plants to be under stress may be due to high temperatures or low lightening, which reduces nutrient usage. At the same, plants keep changing various types of nutrients and nutrient ratios as their rate of growth changes. Due to these reasons, continuous monitoring is needed to check plant nutrient absorption and EC levels. Our developed system can function in both monitoring and dosing plants. The system uses sensors to regularly measure the nutrient reservoir's pH, EC levels, and temperature ranges. And we are using the Growth Stage Identification algorithm to identify the plant's growth stage in its life cycle. Hydroponic systems depend on an exact ratio of micronutrients, phosphorus, potassium, and nitrogen, with suggested amounts based on the crop being grown. Real-time data from sensors measuring EC, pH, and nutrient concentrations in the nutrient solution is interpreted by the automatic dosing system. The system dynamically modifies the dosage of nutrient solutions to maintain them within the

preset optimal ranges in response to these readings. In addition to doing away with the necessity for manual adjustments, this automated and responsive method guarantees that plants get the exact nutrients they need at every stage of growth, encouraging healthier plants and optimizing crop yield in hydroponic farming.

### Growth stage identification algorithm

A subset of machine learning is deep learning. Deep understanding is used to analyze the input image layer-by-layer, with each layer providing more detailed information about the input.

We fed photos of tomato plants photographed into this system for stage identification. Pixels in a rectangular grid make up the given image. The second layer comprehends the image's edges. The following layer builds nodes from the edges. The following would then discover branches from the nodes. The output layer will then find the entire item. This algorithm used multilayer perception and CNN to identify the stage of the tomato plant with high accuracy.

### Mask R-CNN-based growth stage identification

A tomato plant typically shows green leaves with variable sizes and stems and randomly arranged and intricately overlapping leaves. The characteristics extracted from the shallow and deep layers are combined using a feature pyramid network (FPN), and feature mapping improves recognition. When paired with the FPN network, ResNet-50 also acts as the support network for feature extraction. Regarding the loss function, the training loss of the Mask R-CNN comprises two components: the training loss in the multibranch prediction network and the training loss in the region proposal network (RPN). The following is the entire training loss $L_{final}$ calculation formula:

$$L_{final} = L_{RPN} + L_{Mul-Branch} \tag{4}$$

where $L_{RPN}$ consists of the rectangular box regression loss and the anchor classification loss (SoftMax loss) (smooth $L_1$ loss). The $L_{RPN}$ is determined as surveys:

$$L_{RPN} = \frac{1}{N_{cls1}} \sum_i L_{cls}(p_i, p_i^*) + \lambda_1 \frac{1}{N_{reg1}} \sum_i p_i^* L_{reg}\left(t_i, t_i^*\right) \tag{5}$$

where $L_{Mul-Branch}$ is the total of the three branch losses in the multitask prediction network (SoftMax loss, smoothing $L_1$ loss, and mask loss):

$$\begin{aligned} L_{Mul-Branch} &= L\left(p_i, p_i^*, t_i, t_i^*, s_i, s_i^*\right) \\ &= \frac{1}{N_{cls2}} \sum_i L_{cls}(p_i, p_i^*) + \lambda_2 \frac{1}{N_{reg2}} \sum_i p_i^* L_{reg}\left(t_i, t_i^*\right) + \gamma_2 \\ &\quad \frac{1}{N_{mask}} \sum_i L_{mask}\left(s_i, s_i^*\right) \end{aligned} \tag{6}$$

The constant $N_*$ Eqs. (3) and (4) denote the quantity of equivalent anchor points or rectangular boxes. The composed rectangular box regression loss and mask loss,

correspondingly, are hyperparameters $\lambda^*$ and $\gamma^*$. The following formulas are used to derive the classification loss $L_{cls}$, regression loss $L_{reg}$, and mask loss $L_{mask}$:

$$L_{cls}(p_i, p_i^*, p_i) = -\log p_i^* p_i \tag{7}$$

$$L_{reg}(t_i, t_i^*) = smooth_{L1}(t_i^*, t_i) \tag{8}$$

$$smooth_{L1}(x) = \begin{cases} 0.5x^2 & if\ |x| < 1 \\ |x| - 0.5 & otherwise \end{cases} \tag{9}$$

$$L_{mask}(s_i, s_i^*) = -(s_i^* \log(s_i) + (1 - s_i^*) \log(1 - s_i)) \tag{10}$$

where $p_i$ is the ground-truth label probability of anchor i and $p_i^*$ is the classification probability of anchor i. Using four constraint vectors—the width, height, and horizontal and vertical organizes of the points in the rectangle—the moveable $t_i$ reflects the modification among the predicted rectangle box and the ground-truth label box. Additionally, $s_i$ and $s_i^*$ represent the mask binary conditions from the prediction and ground truth labels, respectively, and $t_i^*$ denotes the difference between the ground-truth label box and the positive anchor.

## Procedure for nutrient estimator with visual representation

The Arduino board serves as the main controller of a hydroponic dosing system, coordinating the accurate dispensing of nutrients according to the surrounding conditions. Its central component is the microcontroller, usually an ATmega328p, which runs the programming that defines the logic of the dosing system. Antenna pins interact with sensors that measure factors like pH or nutrition levels, while digital pins are set as outputs to control dosing components, including pumps or valves. Both types of pins are crucial. For reliable performance, the voltage regulator guarantees a steady power supply, and the crystal oscillator gives precise timing for the microcontroller's functions. Programming and interacting with external devices are made easier *via* the USB interface. Programmed logic, in which the microcontroller interprets sensor inputs, decides on the basis of specified criteria, and outputs signals to control dosing components, is what makes the dosing system work. The Arduino board functions as a flexible and effective controller thanks to the dynamic interplay of its parts, guaranteeing the accurate and automated distribution of nutrients in hydroponic farming.

The automatic dosing system consists of an Arduino board with a DHT11 temperature sensor, pH, and EC meter. We use a water tank filled with RO water, as in the deep-water culture model, with an aerator supplying oxygen for roots. We must start the system with the required pH range between 5.5 to 6.5 and EC values between 2 to 5. We used submersible pumps for pumping water and nutrients. The amount of time the PUMP is ON or OFF depends on the stage of the plant, pump speed, tank volume, and the number of plants. The programmed Arduino microcontroller periodically records updated values and adjusts the system with pH and EC as per thresholds. It captures the image of the plants to identify the current growth stage at regular intervals. It then stores all the measured data in the IOTCloudData web server that helps the farmer to feed plants with

the right amount of nutrients and supplements. Figures 3 and 4 show the flowchart of the nutrient hydroponic automatic controlling system. We programmed our dosing system to adjust between ratios of nutrients for different stages in plant development. Suppose the system is unsuitable for plants with violated pH or EC values. The controller immediately responds and activates an appropriate relay to correct pH or EC to the preferred values. Total four relays with four pumps

- Pump 1 with Relay 1 to fill the water tank
- Pump 2 with Relay 2 to pump nutrients A
- Pump 3 with Relay 3 to pump nutrients B
- Pump 4 with Relay 4 to drain the water tank

The systems adjust 5V DC submersible pumps flowing time using Eq. (2) based on the plant's growth stage.

## FLOWCHART

There are several types of hydroponic systems. Here in this system, deep water culture has been followed for growing plants. This entire setup is designed with a growing medium. This water reservoir is connected with nutrient mixture pumps and an aerator machine to provide oxygen to the bottom of the root. Our hydroponic system is connected to the Arduino microcontroller, which is connected to sensors that sense various features such as temperature, pH, and EC in regular intervals. Every time after sensing, measured values are compared with threshold values already set in the system. Initially, a water level sensor called an ultrasonic sensor is fixed on the microcontroller, checking whether the water is overflowing or underflowing. If the water equal is less than the threshold level, the water pump is on immediately. Once the water level reaches the specified level, it turns off the water pump. If pH is low and less than the pHLow, switch on the water pump to reduce the acidic nature. Otherwise, if pH is more than pHHigh, add some pH corrective solution called citric acid to reduce the basicity. To adjust EC, if it finds EC measured is higher than the ECHigh, water is added to balance the nutrient solution. Otherwise, if the calculated EC value is lower than the threshold range specified, ECLow, then call the Growth Stage Identification algorithm to determine the current plant growth stage and add the nutrient solution for the estimated time. The estimation of nutrients to pump in the system and the amount of time needed to pump the answer is pictorially shown in Fig. 4.

### The application of the growth stage identification algorithm in hydroponic farming

The application of the Growth Stage Identification algorithm in hydroponic farming situations has produced significant improvements in a variety of real-world applications. The precise detection of growth stages by the algorithm led to optimal nutrient administration in a tomato-growing commercial greenhouse, minimizing nutrient waste and increasing yield. By automating the procedures of monitoring and fertilizer correction based on accurate growth stage recognition, the algorithm improved labor productivity in

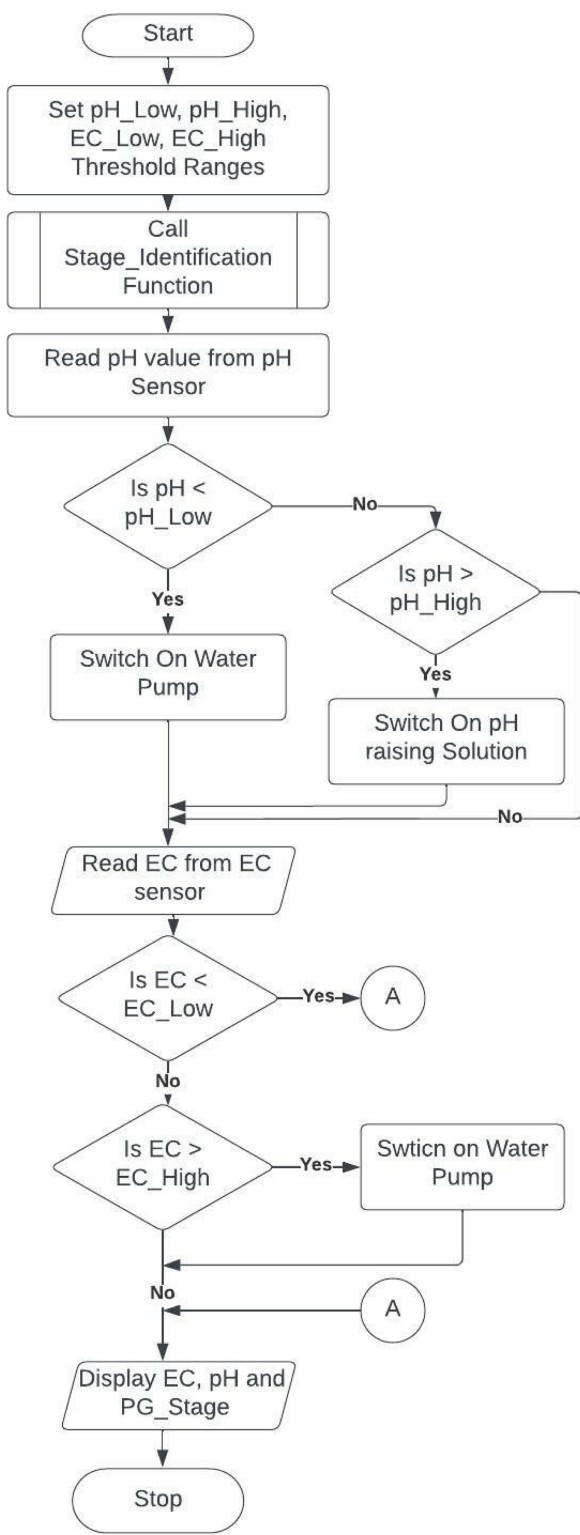

**Figure 3 Main loop to check EC and pH based on the plant stage.**

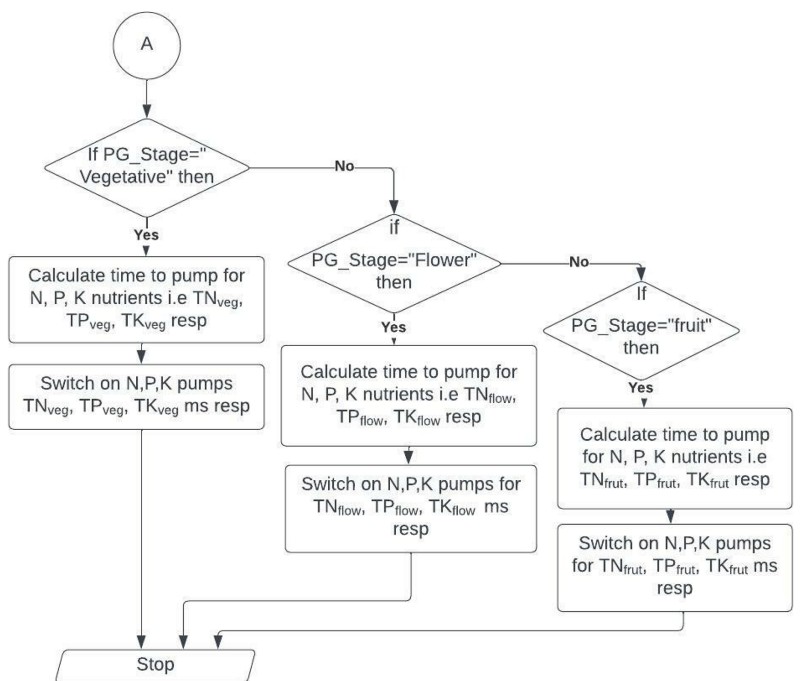

**Figure 4 Flowchart for identifying plant stage.**

vertical farming systems and streamlined operations. Additionally, the algorithm's use in sustainable urban agriculture practices shown how it may reduce environmental impact by optimizing nutrient solutions, satisfying consumer demand for environmentally friendly food. Scholarly establishments applying the algorithm saw enhanced crop quality as a result of precise nutrient distribution, while arid-region hydroponic farms took use of its potential for resource-efficient farming. Furthermore, the algorithm's applicability to automated home hydroponics showed that even city people could use it to produce plants successfully and optimally. All of these case studies highlight how adaptable and powerful the Growth Stage Identification algorithm is for raising production, sustainability, and efficiency in a range of hydroponic farming scenarios.

# EXPERIMENT RESULTS

## System setup

The proposed system has been implemented to estimate the extra nutrients required to feed the plants in the right amounts. Files S2 and S3 show the system implementation in real-time for visualizing plant growth from vegetative to fruit.

The above experiment used deep water culture to grow the plants in a fixed water tank mixed with the initial amount of nutrients. The tomato plants are taken for demo and monitored from the seedling stage to the fruiting stage for nearly 115 days. The plants were in the observation to analyze that nutrient consumption changes as they change their growth stage.

We used an Arduino board designed with a pH sensor, TDs sensors, and a temperature sensor connected to the outside world using ESP-32 Wi-Fi for automating the nutrient supply. The Arduino board is coded using the software Arduino IDE for the above implementation. Its user-friendly open-source software contains a text editor for writing and uploading data to the board. In this software, most of the components are written and validated using JavaScript. The board stores all the data logs processed and communicates with a private IOTCloudData server connected with an integrated ESP-32 Wi-Fi module.

## Plant growth identification

In this system, we automated nutrient supply according to the plant stage. This idea was implemented using the Plant Growth Identification algorithm to record the plant's maturity. The experimental results of the growth stage identification method showed that it was very accurate and reliable. An impressive 97.5% of the time, the program correctly identified the vegetative stage, showing that it is good at telling the difference between early growth stages. Furthermore, the accuracy stayed high during the flowering stage at 97.5%, showing that the algorithm is reliable at picking up on small changes in plant growth. It was during the fruiting stage that the accuracy rate was the highest, hitting an amazing 99.4%. This exceptional accuracy shows how well the algorithm can exactly pinpoint the complex physiological changes that happen during fruit development. Correlations between the algorithm's accuracy rates and the nutrient solutions' parts per million (ppm) values further proved that it was reliable. This showed that accurate growth stage identification and targeted nutrient delivery in the hydroponic system worked well together. It used USB to RS-232 cable to update data from the laptop to the cloud server IOTCloudData. This algorithm successfully detected the plant stages with 99.9% accuracy. The tomato plant images are captured using a camera at the farming site, and some of the photos are taken from Kaggle datasets for training and testing. We use Python programming language to implement this algorithm. TensorFlow and Keras, and Numpy libraries are used for implementation. The algorithm implementation is shown for identifying vegetative, flowers, and fruit in Files S4–S6, respectively.

## Estimating and automating nutrient supply

The Arduino board connected the water tank with nutrient bottles A and B to supply the estimated amount of nutrients calculated. Based on the data received from sensors, the controller processes the data and sends instructions to the actuator to turn on Nutrient A and Nutrient B pumps for the amount of time calculated using the above equations with input parameters plant stage, tank volume, and surplus amount of nutrients required at that stage.

For example: Assume plants are at the flowering stage, and the measurement of nutrients in the 15-l water tank was reduced to 1,700 ppm. The system then immediately switches on the nutrient 5 V submersible pumps with a rate of 2 LPM to supply adequate amounts at the right time.

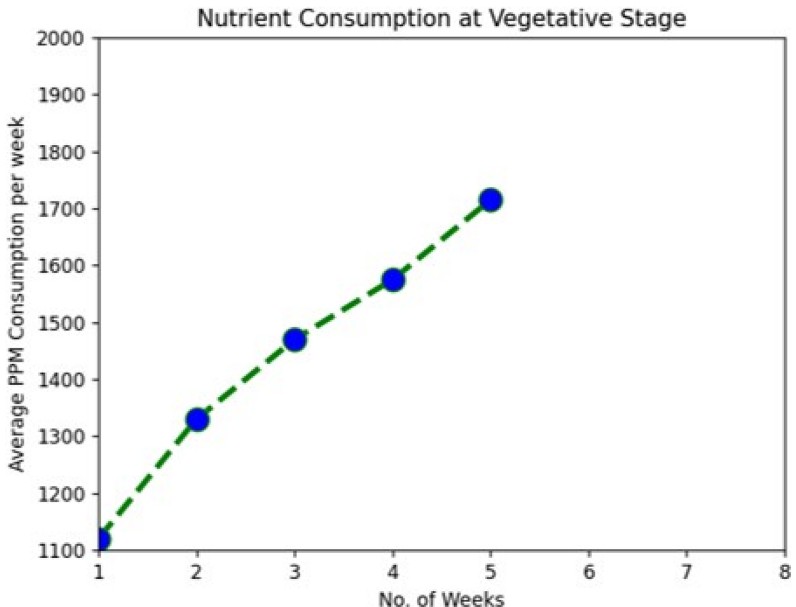

**Figure 5 Nutrient consumption at vegetative stage.**

The amount of nutrient solution needed

$$= \text{MAXFLOW}_{\text{ppm}} - \text{SUR}_{\text{ppm}} \qquad (11)$$
$$= 2,500 - 1,700$$
$$= 800 \text{ ppm (Means 800 mg/Liter)}$$

Then the time needed to keep the pumps ON.

$$= \frac{15 \ Liter \ \times \ 800 \ mg/Liter}{2 \ \times \ 10^6 mg/Minute} \qquad (12)$$

(Sub in Eq. (11))

$$= 0.36 \ sec \ (1 \ \text{Liter} = 1,000,000 \ \text{mg})$$

## Data analysis for nutrient supply

During the system implementation period, it was observed that the plants were taking nutrients at increasing levels as they were growing from one stage to another. We used Tomato plants for the demo and were kept in observation for around 115 days from the installation. With a view of data downloaded from the cloud server, we have clearly shown the average nutrient consumption per week in Table 2 and Figs. 5–7.

The following graph analysis developed using Python with the Matplotlib Library clearly shows how plant nutrient consumption rate increase with change in the growth stages.

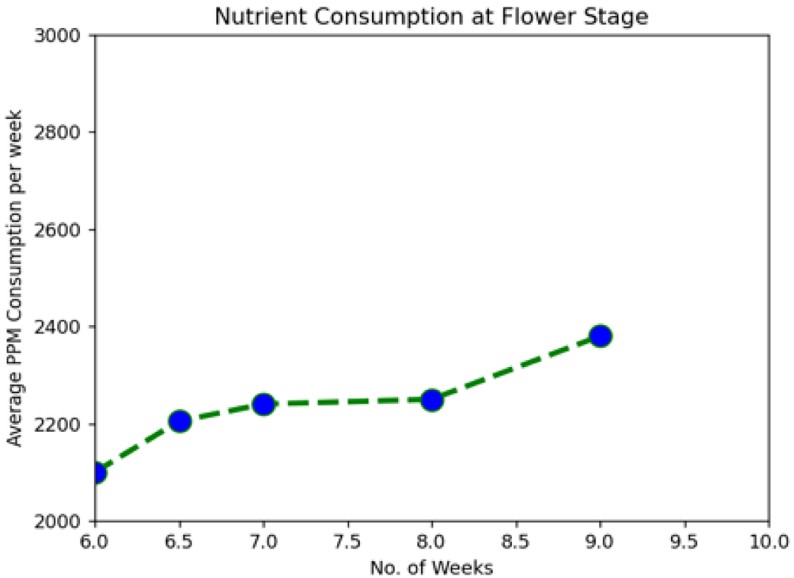

**Figure 6 Nutrient consumption at flower stage.**

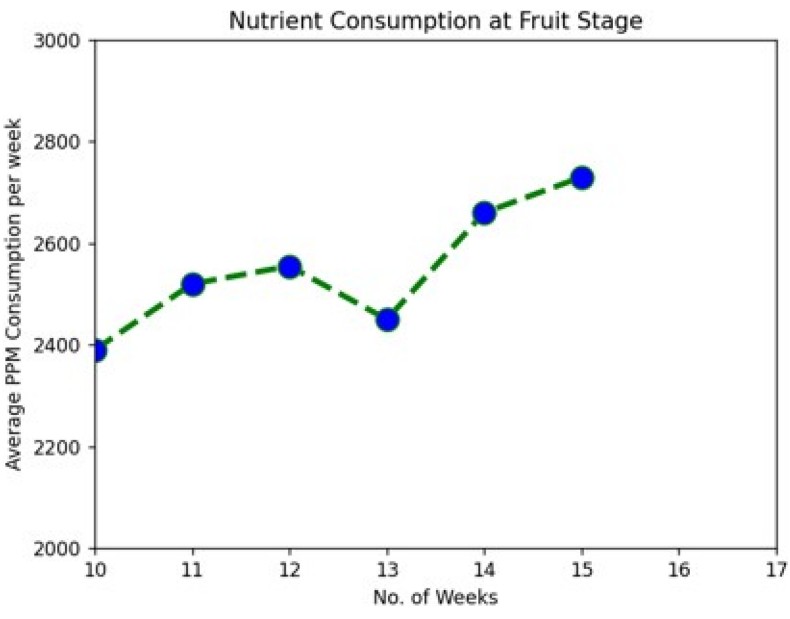

**Figure 7 Nutrient consumption at fruit stage.**

## Accuracy analysis chart for comparing the proposed with the existing system

Figure 8 and Table 3 compare the accuracy of the proposed methodology to various current methods. The graph shows that the deep learning approach improves performance and accuracy. For example, in the 1–5th week, the proposed model has a 97.5% accuracy value, while the CNN (*Gul & Bora, 2023*) and DNN (*Arvind et al., 2020*) models have 95.5% and 94.3% accuracy values, respectively. The proposed model, on the other hand,

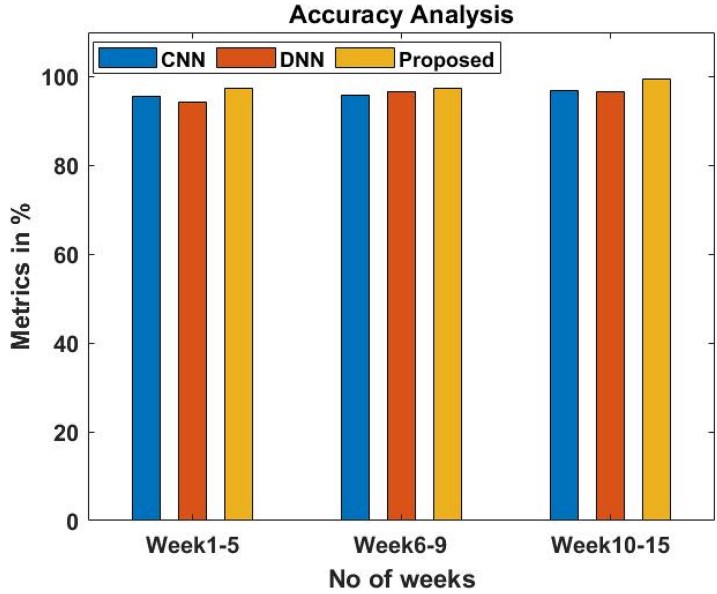

**Figure 8 Accuracy analysis of the proposed system.**

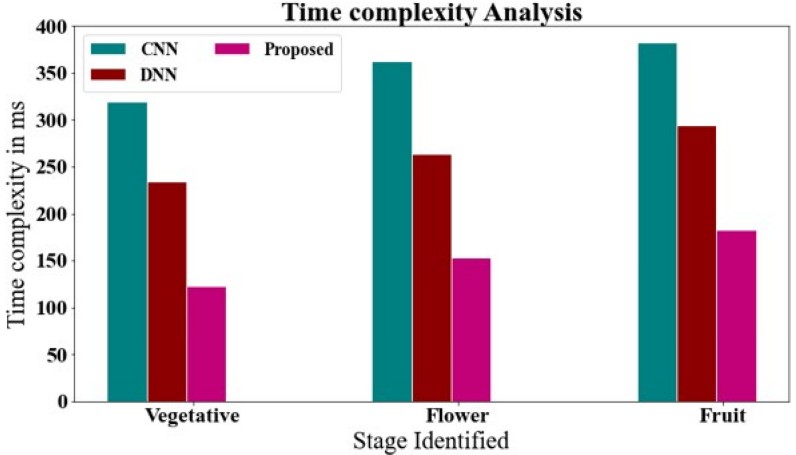

**Figure 9 Time complexity analysis of chart for comparing the proposed with the existing system.**

fared better with varied weeks. Similarly, the proposed model has an accuracy value of 99.4% under 10–15 weeks, whereas CNN and DNN have 96.8% and 96.6%, respectively.

## Time complexity analysis of chart for comparing the proposed with the existing system

Figure 9 and Table 4 show the time complexity of the proposed methodology is compared to that of existing techniques. The data clearly shows that the proposed technique outperformed all other strategies. The proposed approach, for example, took only 123.23 ms to with vegetative stage, whereas other existing methods such as CNN and DNN have taken 319.24 and 233.83 ms, respectively. Likewise, the proposed approach takes 152.82 ms

**Table 4 Time complexity analysis of chart for comparing the proposed with the existing system.**

| Week number | Stage identified | CNN | DNN | Proposed |
|---|---|---|---|---|
| Week 1–5 | Vegetative | 319.24 | 233.83 | 123.23 |
| Week 6–9 | Flower | 362.84 | 263.73 | 152.82 |
| Week 10–15 | Fruit | 382.64 | 293.64 | 182.54 |

with flower stage, while existing techniques like CNN and DNN take 362.84, and 263.73 ms, respectively. Similarly, the proposed approach takes 182.54 ms with fruit stage, while existing techniques like CNN and DNN take 382.64, and 293.64 ms, respectively.

## CONCLUSION

Hydroponics makes people more comfortable growing their organic and pesticide-free food quickly at home. The main issue with hydroponics is the constant monitoring of the pH and EC levels of the nutrient solution, as well as the surrounding temperature and humidity range. Hydroponics culture highly depends on human attention to supply the suitable parameters for the quality growth of plants. In this process, it was observed that plants nutrients consumption changed based on the plant stages such as germination, vegetative, reproduction, and harvesting. The number of nutrients the plants consume is different in all locations. It is required more focus on identifying plant stage and automating the distribution of nutrients accordingly. In this research, we proposed a Growth Stage Identification algorithm to extract plant growth and avoid undersupply or oversupply of nutrients. The experimental results showed that the Growth Stage Identification algorithm achieves a 97.5% accuracy rate for the first 5 weeks with 1,715 ppm of nutrition consumption which identify the vegetative state. A total of 97.5% accuracy rate for the 6–9th week with 2,380 ppm of nutrition consumption, which identify the flowering stage, and 99.4% accuracy rate for the last 10–15th week 2,730 ppm of nutrition consumption, which determine the fruiting location of the plant. In the future, integrating machine learning for area computation of the complete plant bush and real-time monitoring seems like a really promising improvement. Utilizing sophisticated algorithms and real-time video feeds, the system provides current data on plant development and well-being, enabling informed choices and accurate system operation. It demonstrates the advantages of merging robotics, IoT, and data analysis for optimal plant cultivation and agricultural practice optimization.

### Funding
The authors received no funding for this work.

### Competing Interests
The authors declare that they have no competing interests.

## Author Contributions

- Tupili Sangeetha conceived and designed the experiments, performed the experiments, analyzed the data, performed the computation work, prepared figures and/or tables, authored or reviewed drafts of the article, and approved the final draft.
- Ezhumalai Periyathambi analyzed the data, authored or reviewed drafts of the article, and approved the final draft.

## Data Availability

The raw measurements are available in the Supplemental Files.

## Supplemental Information

Supplemental information for this article can be found online at http://dx.doi.org/10.7717/peerj-cs.1871#supplemental-information.

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
