# Peer review of "Automatic nutrient estimator: distributing nutrient solution in hydroponic plants based on plant growth"

_PeerJ Computer Science, doi:10.7717/peerj-cs.1871_

## Round 0.1 · original submission · Major Revisions

The paper must be improved according to the reviewers' suggestions.

Reviewer 1 ·

Basic reporting

The introduction provides a clear overview of the topic, but it could benefit from more context on the importance and challenges of nutrient distribution in hydroponic systems. Providing a brief review of relevant literature and emphasizing the existing gaps would enhance the introduction.

Experimental design

Author described the experimental setup, including the number of plants, environmental conditions, and duration of the experiments well. Also they explained selection of specific plant species or varieties, as different plants may have varying nutrient requirements.

Validity of the findings

Author presented the article entitled"Automatic nutrient estimator: Distributing nutrient 1 solution in hydroponic plants based on plant growth" with good experimental setup. Hardware implementation is appreciable with quantitative methods. However here some suggestions to be incorporated in the revised manuscript to improve the quality of this manuscript.
Abstract section may be changed by updating the methods importance and objectives to overcome the existing challenges in "Distributing nutrient solution in hydroponic plants based on plant growth".
The final outcome of the results may be included in the abstract sections.

Also author may include the novelty of this Distributing nutrient solution in hydroponic plants in survey sections. How they claim the proposed model achieving the best accuracy than existing CNN and DNN model ?.

Recent survey may be discussed with the outcome of the best model and its limitations in the survey sections. based on these author can identify the problems and claim the proposed model novelty of this research

What is NPK ? usually Nitrogen (N), Phosphorus (P), and Potassium (K) are primary nutrients. Author means this one as NPK ?

How the proposed model is performing in hydroponic plants ? what is the optimization techniques here ? so section 3 is weak. Author can include more clear descriptions of proposed works.

Final statistical analysis may be improved by adding time complexity analysis and testing validation or trining validation of the models.

Conclusion section is not sufficient to conclude the research. Author may described the final outcome of this proposed model and how its benefitted for future research etc.....

Annotated reviews are not available for download in order to protect the identity of reviewers who chose to remain anonymous.

Reviewer 2 ·

Basic reporting

The report effectively highlights the importance of adjusting nutrient delivery based on the plant's growth stage. The incorporation of sensors and an identification algorithm to achieve this is a strong point in the methodology. In this article, sufficient reference materials are included.

Experimental design

The introduction provides a clear explanation of the problem, the relevance of hydroponic cultivation, and the need for effective nutrient management. It effectively sets the stage for the experimental design.

Validity of the findings

The methods section is logically connected to the problem statement. It explains how the proposed automatic hydroponic nutrient estimator addresses the problem of nutrient management by considering the plant's growth stage.

Additional comments

The author should address the below comments for further processing
1. How does the system ensure that nutrient quantities are supplied according to the specific growth stage or age of the plants, and what parameters are monitored to achieve this?
2. What measures are taken by the system if the recorded pH and EC values fall outside predefined ranges, and how does it adjust the nutrient concentration to maintain optimal levels?
3. Could you provide more details about the Growth Stage Identification algorithm and how it determines the plant's age to dispense the correct quantity of nutrients?
4. How does the system interact with the farmer through the View atSite app and IoTCloudData cloud server, and what information is transmitted to the farmer's mobile device?
5. How do you ensure that the automatic dosing system maintains nutrient levels within the recommended thresholds, and how are these thresholds determined?
6. Can you provide more details about the specific components of the Arduino board and how they interact to control the dosing system?
7. How do you envision the practical application and scalability of this system in real-world hydroponic farming scenarios, and what are the potential benefits for farmers and the environment?
8. What are the implications of achieving a 97.5% accuracy rate for identifying the vegetative stage, a 97.5% accuracy rate for the flowering stage, and a 99.4% accuracy rate for the fruiting stage with the proposed algorithm?
9. Can you explain the experimental results related to the accuracy of the Growth Stage Identification algorithm, including the specific accuracy rates and ppm values for different plant stages?
10. Are there any practical applications or case studies that demonstrate the real-world benefits of the Growth Stage Identification algorithm in hydroponic farming scenarios?

---

## Round 0.2 · Minor Revisions

Minor revision as suggested the reviewers.

Reviewer 1 ·

Basic reporting

Author revised the manuscript "Automatic Nutrient Estimator: Distributing nutrient solution in hydroponic plants based on plant growth" well. However i need some more minor changes in the following areas.
The methods which the author was developed was described very lengthy in abstract. So better author can reduce the description in abstract.
Final outcomes should be explained as result section of abstract instead of background sections.
In introduction section, author explained main benefits from reference[7], this should be changed as paragraph format instead of key points. Also the contents were referred from reference [5] to [7] started as "this method" , so author can specify the method name while describing in introduction section.

Experimental design

Nil

Validity of the findings

Fig 4 and 5 showing that normal flow conventional of proposed system. However this figures are not playing major role in the manuscript. So better author can remove these figures and its content since the basic flow works were already explained well in the introduction and survey section.
Similarly Fig 8, 9, 10, 11, 12 which shows about the plant growth and these images can be moved to the Annexure part of the article. Author can reduce the content from conclusion section since its vague to understand the final outcome

Additional comments

Author can do proofread before submit revision.

Reviewer 2 ·

Basic reporting

The report adeptly emphasizes the need to tailor nutrient supply according to the plant's growth phase, with a notable strength lying in utilizing sensors and an identification algorithm. Additionally, ample reference materials are provided within this article.

Experimental design

The introduction provides a clear explanation of the problem, the relevance of hydroponic cultivation, and the need for effective nutrient management. It effectively sets the stage for the experimental design.

Validity of the findings

The methods section cohesively links to the problem statement, elucidating how the suggested automatic hydroponic nutrient estimator tackles nutrient management concerning the plant's growth stage.

Additional comments

1. Check the author's affiliation, the first author in the journal portal and pdf documents seem different.
2. In Section 3.2, Both the text and Fig 3, Is this diagram needed in the main text, and there is any supplementary section you can move it to?
3. In the reference section, verify whether the [2] is complete or incomplete.

---

## Round 0.3 · accepted · Accept

Te paper was well improved.